# SCENARIO-BASED QUESTION ANSWERING WITH INTERACTING CONTEXTUAL PROPERTIES

**Haitian Sun**
School of Computer Science
Carnegie Mellon University
haitians@cs.cmu.edu

**William W. Cohen**
Google Brain
wcohen@google.com

**Ruslan Salakhutdinov**
School of Computer Science
Carnegie Mellon University
rsalakhu@cs.cmu.edu

## ABSTRACT

In the scenario-based Question Answering (QA) task, models are asked to find answers that are appropriate to the user scenarios associated with the question and identify information that is missing from the scenarios but is necessary for the answers to hold. Scenarios commonly include multiple properties of users, such as age, employment status, and income level for the question "How much can I claim from this benefit". The properties relevant to a potential answer are given in a document, which will state *conditions* necessary for the answer to hold. Documents also may specify how conditions interact with each other, e.g. with text like "one of the conditions below must apply". Although understanding the relationship between conditions is crucial for solving this challenging QA task, limited work has been done so far in modeling this. In this paper, we propose the T-Reasoner model, which solves this problem with three jointly learned modules: an entailment module which checks whether a condition has been satisfied by the scenario, a decoding module which locates eligible answers from documents, and a reasoning module which infers the relationship between conditions and performs a reasoning step to determine the logically consistent answers and identify missing conditions. T-Reasoner outperforms strong baselines on a synthetic scenario-based QA dataset and achieves a new state-of-the-art on two scenario-based QA benchmarks, outperforming the prior best models by 3-10 points. [1]

## 1 INTRODUCTION

Many questions can only be answered correctly after some *context* for the question is supplied or inferred: e.g., "When is the next LA Lakers home game" needs temporal context, and "Where is the closest pizza place" needs geographical context. Prior work on contextual QA (Zhang & Choi, 2021; Dhingra et al., 2021; Kasai et al., 2022; Chen et al., 2021) has focused on tasks in which context is important, but limited: generally a small number of properties of the user that posed the question need be considered (e.g., location and time). However, many important questions depend on many more properties of the user. In this paper we consider scenario-based QA, in which questions are augmented with a textual "scenario" that describes some properties of the user. For example, in Figure 1 a user has posed a question "how much support am I eligible for?" , and the answer depends on multiple user properties (namely, their relationship with deceased, and whether they or other relatives have claimed other benefits.) Having multiple contextual properties means these properties can interact. For example, in Figure 1 the answer depends on a conjunction of conditions (e.g. "if both" in Scenario 1) and also a disjunction of conditions (e.g. either being a "relative" or a "close friend" in Scenario 2).

In our benchmarks, scenarios are informative but not complete, so the goal of the system is to identify possible answers—i.e., answers that are *logically consistent* with the scenario—as well as any conditions that necessary for the answer to hold which are *not* entailed by the scenario. For example, in Figure 1 Scenario 1, the system should provide the answer "up to $1200" but must also note that the condition "you didn't claim other benefits" is required by the answer, and not entailed by the scenario. We refer to such conditions as *unsatisfied conditions*. This task is challenging because in addition to finding eligible answers from documents, it also requires models to perform two non-trivial reasoning tasks. First, it must understand the document well enough to understand conditions given as

---

[1] Codes and data are available at https://github.com/haitian-sun/T-Reasoner.

context for the answer (each property that may affect the answer is considered as a condition), and the logical relationship between these conditions. For example, in Figure 1 Scenario 1, it requires *both* "the partner of the deceased..." and "you didn't claim other benefits" to be satisfied (i.e. conjunction), while it requires *either* a "relative" *or* "close friend" (i.e. disjunction) in Scenario 2. Second, a model must identify which conditions are entailed by information provided in user scenarios, which are contradicted, and which are not mentioned but are required to support an eligible answer.

Previous work by Clark et al. (2020b) has shown that pretrained Language Models (LMs), e.g. RoBERTa (Liu et al., 2019), can be finetuned to perform a similar reasoning task over hypothetical statements, i.e. "if A and B then C". However, conditions used in their experiments are over simplified and sometimes semantically incorrect, e.g. A = "Mike is strong" and B = "Cindy is green". Furthermore, languages used to described the relationship between conditions are easy, and the number of conditions involved in the reasoning process is small. All factors above make the proposed task easy for existing models (Liu et al., 2019; Raffel et al., 2019), but under-represents the challenges exists in real problems that require reasoning with logically interacting conditions.

Furthermore, previous work (Clark et al., 2020b) makes an assumption that every conditions must be either satisfied or contradicted by the evidence provided in questions. As a result, no "unsatisfied condition" is required in predictions. We do not make such assumption, but instead only provide evidences for a subset of conditions, and ask models to predict a logically consistent answer and identify conditions that are required but not yet satisfied, i.e. unsatisfied conditions. Indeed, experiments (Sun et al., 2021a) show that pretrained language models (LMs), e.g. T5 (Raffel et al., 2019), struggle to predict unsatisfied conditions. Even though an additional module is specifically trained to predict unsatisfied conditions (Gao et al., 2020b; Ouyang et al., 2020), their performance is still limited.

We propose a simple yet effective model, T-Reasoner, which models the relationship between conditions and performs the reasoning task to verify answers that are consistent with user scenarios and identify conditions that are unsatisfied. T-Reasoner contains three main modules, an entailment module, a reasoning module, and a decoding module, which are jointly trained. The entailment module predicts whether conditions have been entailed or contradicted by users' scenarios. The reasoning module infers the relationship between conditions then performs a reasoning step to decide whether the provided information in user scenarios is sufficient and to identify unsatisfied conditions otherwise. If the answer is a free-form text span, T-Reasoner additionally uses a generation module to predict the answer span. T-Reasoner shows excellent reasoning ability on a synthetic dataset and outperforms the previous state-of-the-art models on two Question Answering (QA) datasets, ConditionalQA and ShARC (Sun et al., 2021a; Saeidi et al., 2018), improving the state-of-the-art by 3-10 points on answer and unsatisfied condition prediction tasks.

## 2 RELATED WORK

The task proposed by Clark et al. (2020b) is commonly referred to as *deductive reasoning* where all information required to find a definite answer is provided. Other models have been developed for deductive reasoning with symbolic rules (Cohen, 2016; Cohen et al., 2020; Sun et al., 2020; Ren et al., 2020; Ren & Leskovec, 2020). Embedding-based methods (Sun et al., 2020; Ren et al., 2020; Ren & Leskovec, 2020) first convert symbolic facts and rules to embeddings and then apply neural network layers on top to softly predict answers. These models differ from our work in that the symbolic structure of the rules is typically known, whereas in our model it is implicit in a document.

Other recent work in deductive reasoning focused on tasks where rules and facts are expressed in natural language (Talmor et al., 2020; Saeed et al., 2021; Clark et al., 2020b; Kassner et al., 2020). Such tasks are more challenging because the model has to first understand the logic described in the natural language sentences before performing logical reasoning. Many of these models rely on rules that are produced by templates, or templated rules that have been paraphrased by crowd workers. In our work, the logical interactions analogous to these rules are implicit in real-world documents.

Different from most reasoning tasks, the task considered in this paper provides a list of conditions that, if true, would support an answer. Identifying such conditions is usually called *abductive* reasoning, as opposed to deductive reasoning. Very limited work has explored abductive reasoning for QA. Previous work (Gao et al., 2020a;b; Ouyang et al., 2020) on the ShARC (Saeidi et al., 2018) dataset propose to solve this problem by predicting a special label "inquire" if there was not enough information to make a definite prediction. Specifically, EMT and DISCERN (Gao et al., 2020a;b) computed

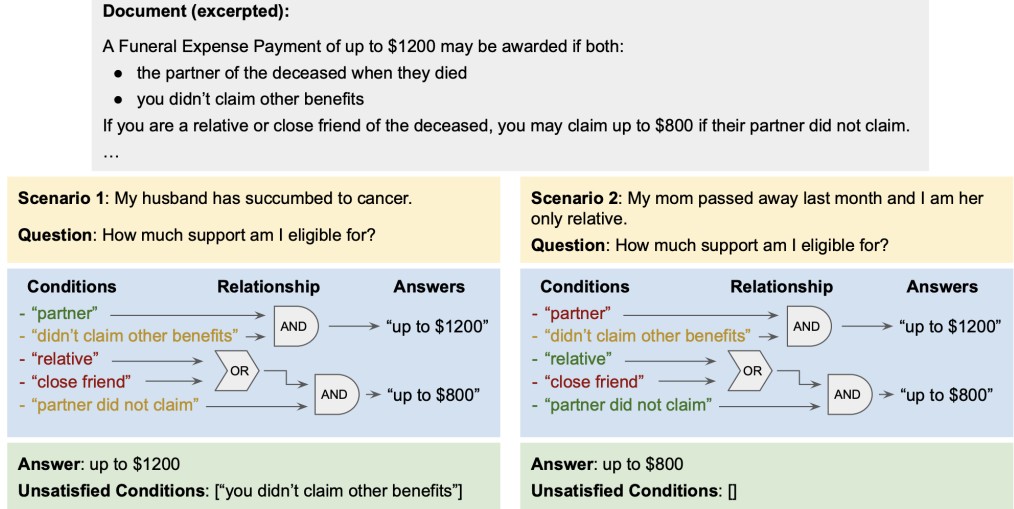

Figure 1: Examples for the scenario-based QA task. The two examples have the same question but different scenarios which lead to different answers. Diagrams in the blue blocks show the desired reasoning process. The conditions in green, red, and yellow are entailed, contradicted, and not mentioned in the scenario. In Scenario 1, the answer "up to $1200" is logically consistent with the scenario. The condition "you didn't claim other benefits" is an unsatisfied condition because it is not mentioned but required to be satisfied for the answer. The other condition "partner did not claim" is also not mentioned but is not an unsatisfied condition because whether it is satisfied will not affect the answer. In Scenario 2, there's not an unsatisfied condition because existing information provided in the scenario is sufficient to predict the answer "up to $800".

an entailment vector for each condition and performed a weighted sum of those vectors to predict the final answer. DGM (Ouyang et al., 2020) additionally introduced a GCN-based model to better represent the entailment vectors. Even though these models were able to predict the answer labels as "inquire" when there were unsatisfied conditions, none of them predict *which* conditions needed to be further satisfied, unlike our model. Our model is also more scalable than these, as it does not require concatenating a full context and a question.

## 3 MODEL

### 3.1 TASK: QA WITH CONDITIONS

The scenario-based QA task requires models to find answers that are logically consistent with the provided user scenarios which are potentially incomplete. In this paper, we consider this task in the reading comprehension (RC) setting in which a passage that contains relevant information about the question is provided. We leave the open-domain setting of this problem for future work. Specifically, a model takes a question, a scenario, and a passage that contains answers and conditions as input and predicts logically consistent answers and their unsatisfied conditions. Let's consider a passage that contains a set of conditions $C = \{c_1, \ldots, c_n\}$ and the set of eligible answers for a question under all possible combinations of conditions $A = \{a_1, \ldots, a_m\}$. Each answer $a_i \in A$ is restricted by a subset of conditions $C_i \subseteq C$. Conditions in $C_i$ interact with each other under relationship $R_i$ ($R_i$ is an abstract set which will not be explicitly expressed). A condition group, $G_i = (C_i, R_i)$ is a pair of $C_i$ and $R_i$, which describes in what scenario the answer $a_i$ is correct. Note that the list of answers $A$, sets of conditions $C_i$'s and their relationship $R_i$'s are not explicitly provided in training and testing examples – models have to generate them from the passage.

We say that a condition group $G_i$ is *satisfied* if its underlying logical statement that consists of $C_i$ and $R_i$ has been satisfied by the scenario, for example, in Scenario 2 in Figure 1 where the condition group for "up to $800" has been satisfied. Besides being satisfied, a condition group $G_i$ has two more possible outcomes: (1) $G$ is *partially satisfied* if some of the conditions have been satisfied but there is still some information missing so the answer is not fully supported, e.g. the condition group of the answer "up to $1200" in Scenario 1 (Figure 1), and (2) $G_i$ is *contradicted* if one or more

conditions in the group are contradicted which causes the answer ineligible, e.g. the condition group of the answer "up to $1200" in Scenario 2 (Figure 1).

An answer $a_i$ is logically consistent with the scenario if the underlying condition group $G_i$ is satisfied or partially satisfied. We denote the set of logically consistent answers $\tilde{A} \subseteq A$. The set $\tilde{A}$ contains zero or more answers – the set $\tilde{A}$ is empty if none of the answers in $A$ is logically consistent with the user scenario. A model should predict an answer from $\tilde{A}$ if $\tilde{A}$ is not empty, and mark the question as not answerable, otherwise.[2] In addition to predicting logically consistent answers, we also perform the task of finding unsatisfied conditions $\tilde{C}_i$. The set $\tilde{C}_i$ should be concise, i.e. it should only include the conditions that are necessary. For example, the condition "have worked for more than 4 years" is not an unsatisfied condition because whether it has been satisfied or not won't affect the output of the condition group.

In summary, we evaluate a model's prediction of a logically consistent answer $a_i \in \tilde{A}$ and the set of unsatisfied conditions $\tilde{C}_i$ for answer $a_i$, i.e. $(a_i, \tilde{C}_i)$. Answers and unsatisfied conditions in the output are jointly evaluated.[3] This task specifically challenges models' ability in understanding the relationship between conditions and performing logical reasoning process accordingly. We will introduce a simple and effective model, T-Reasoner, to tackle this challenging reasoning task.

## 3.2 MODEL

In this section, we will discuss T-Reasoner which consists of an entailment module, a reasoning module, and optionally a decoding module, to perform this challenging QA task in embedding space.

**Input** The model, T-Reasoner, takes a question $q$ with scenario $e$ and a passage $p$ as inputs and predicts an answer $a_i$ that is logically consistent with the user scenario and a list of unsatisfied conditions $\tilde{C}_i$. Since the list of all conditions $C$ for the question are not provided in the example, we chunk the passage $p$ into pieces and consider each piece of text as a condition $c_i$. Conditions obtained this way may be irrelevant to questions. We rely on the entailment module (see next) to decide whether a condition $c_i$ is relevant and what is its relationship with others. The chunking strategy may be different for different datasets. Please see §4.2 and §4.3 for more information. Briefly, passages are usually chunked into sentences, short passages with 2-3 sentences, or sub-sentences (text phrases).

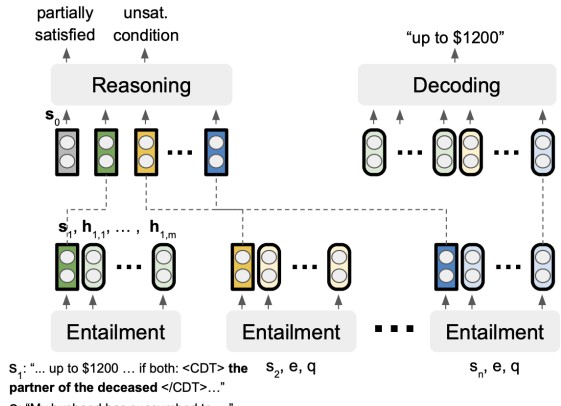

Figure 2: **T-Reasoner Overview**. The entailment module independently encodes each condition $c_i$ (along with its contextual information $s_i$). The entailment module outputs a condition embedding $\mathbf{s}_i$ that will be input into the reasoning module to decide if conditions have been satisfied and identify unsatisfied conditions. Other computed embeddings $\mathbf{h}_{i,j}$ will be used by the decoding module to generate answer spans (if the question has a free-form answer).

**Entailment Module** We apply an entailment module to check whether each condition $c_i \in C$ have been entailed by the user scenario. Each condition $c_i$ is checked independently, as opposed to concatenating all conditions into a long input and checking them all at once. This strategy significantly reduce the computation cost compared to checking all conditions at once, especially if context is long, e.g. legal documents which are tens or hundreds of pages long (see examples in 4.3). Specifically,

---

[2]An oracle model should be able to predict all answers from $\tilde{A}$. We consider a slightly simplified setting in this paper in which a model is only required to predict one of the answers. In our experiments, the ShARC (Saeidi et al., 2018) dataset only contains questions that have a single answer, i.e. $|\tilde{A}| = 1$. The ConditionalQA (Sun et al., 2021a) dataset contains questions that have multiple answers, $|\tilde{A}| > 1$, so the performance will be sacrificed. We leave the task of predicting all logically consistent answers as future work.

[3]Evaluation metrics are different in different datasets Sun et al. (2021a); Saeidi et al. (2018). Please refer to §4.3 and 4.2 for more details.

the computation complexity of our approach is $O(|C|)$ where $|C|$ is the number of total conditions, compared to a complexity of $O(|C|^2)$ otherwise.

This independent checking strategy, however, separates each condition from its context and thus causes a lost of contextual information for conditions $c_i$ and eventually negatively impacts the model's performance. Thus, we extend a condition $c_i$ by adding tokens from its surroundings. For example, the condition "the partner of the deceased when they died" is expanded to "... up to \$1200 may be awarded if both: <CDT> the partner of the deceased when they died <\CDT> you didn't claim ...", where <CDT> and <\CDT> are two special tokens that mark the beginning and end of the condition $c_i$. Apart from making a condition $c_i$ more fluent and coherent, the added contextual tokens also make it easier to understand the relationship between the current condition $c_i$ and other conditions in its neighbours. We may additionally add page titles, section titles, prompts of list items, or table headings, etc., if applicable to the expanded conditions. Please see §4.3 and 4.2 for more details. We denote conditions with extended contextual information as $s_i$ for condition $c_i$.

We learn a Transformer model for the entailment module which takes an expanded condition $s_i$ and the question $q$ and scenario $e$ as input, and returns a list of vectors $\mathbf{s}_i, \mathbf{h}_{i,1}, \ldots, \mathbf{h}_{i,m}$. The first vector $\mathbf{s}_i$ is a summarization vector which includes several aspects of information: (1) whether the underlying condition $c_i$ has been satisfied, contradicted, or not mentioned by the user scenario, (2) whether the condition $c_i$ is relevant to the question, and (3) if relevant, what is its relationship with other conditions in its neighbours. These information will be used for reasoning in the future layers. Embeddings $\mathbf{h}_{i,1}, \ldots, \mathbf{h}_{i,m}$ are token embeddings that will used for decoding if needed. Please see the description of the reasoning module for more information.

$$\mathbf{s}_i, \mathbf{h}_{i,1}, \ldots, \mathbf{h}_{i,m} = \text{Entail}(s_i, e, q) \tag{1}$$

One may consider supervising this entailment module by adding classification layers on $\mathbf{s}_i$ to explicitly predict the entailment status of condition $c_i$ and its relationship with other conditions. However, obtaining supervision labels for these auxiliary tasks can be challenging as they are often not provided in the example. Fortunately, we show that our proposed model, T-Reasoner, can be trained end-to-end, without such intermediate supervision.

**Decoding Module** The decoding module generates an eligible answer $\hat{a}_i$ which is potentially logically consistent to the question. The generated answer $\hat{a}_i$ will not be returned until the status of its condition group $\hat{G}_i$ is verified by the reasoning module (discussed below).

The decoding module is analogous to FiD (Izacard & Grave, 2020), i.e. token embeddings from different conditions (which are encoded separately) are concatenated for decoding. Different from Izacard & Grave (2020) which was applied to independently retrieved passages for open-domain QA, the decoding module in T-Reasoner is used on coherent content, i.e. conditions from the same passage. The contextual information in the expanded condition $s_i$ helps connect conditions that are separately encoded. The decoding module takes token embeddings for all conditions $\mathbf{h}_{1,1}, \ldots, \mathbf{h}_{n,m}$ computed from Eq. 1 to generate answer spans. The generation task is trained with teacher forcing. We do not write out the teacher forcing decoding loss $l_{\text{decode}}$ here. Please refer to the T5 paper (Raffel et al., 2019) for more information. If questions have multiple logically consistent answers, i.e. $\tilde{A} > 1$, we randomly select an answer $a_i \in \tilde{A}$ as the label to train the decoding module.

$$\hat{a}_i = \text{Decode}(\mathbf{h}_{1,1}^{(1)}, \ldots, \mathbf{h}_{k_n,m}^{(n)}) \tag{2}$$

We consider two different types of answers: "Yes"/"No" or free-form answers. In the first case, we simply let the model generate a special a special token [YESNO] and consider the reasoning result from the reasoning module (see next) as the answer, i.e. the answer is "Yes" if the condition group is satisfied (or partially satisfied) or "No" if contradicted. Since some datasets only contain "Yes"/"No" questions, we can then safely discard the decoding module for these datasets. In the second case, i.e. answers are free-form text spans, we will return generated spans as answers only if their condition groups have been verified as satisfied or partially satisfied by the reasoning module. If the condition group is contradicted, we will mark the question as not answerable.

**Reasoning Module** The reasoning module combines the local relationship between conditions from their embeddings $\mathbf{s}_1, \ldots, \mathbf{s}_n$ and performs a logical reasoning process to decide the reasoning result for a condition group $G_i$ for the generated answer $\hat{a}_i$ and to identify unsatisfied conditions $\tilde{C}_i$.

The input to the reasoning module is a list of vectors, $\mathbf{s}_1, \ldots, \mathbf{s}_n$ for conditions $c_1, \ldots, c_n$, that are output by the entailment module (Eq. 1). We use another Transformer model as our reasoner, because

Transformers have the self attention mechanism which allows conditions $\{s_1, \ldots, s_n\}$ to attend to each other, so the reasoning result of a condition group can be summarized. This is crucial because, for example, if one of the conditions in a disjunction group is satisfied, the condition group will be automatically satisfied regardless the status of other conditions in the same group. We prepend a trainable vector $\mathbf{s}_0$ to the list of condition embeddings to summarize the reasoning result.

The output vectors $\hat{\mathbf{s}}_0, \hat{\mathbf{s}}_1, \ldots, \hat{\mathbf{s}}_n$ will be used to predict the status of the condition group and the unsatisfied conditions for the generated answer. The first vector $\hat{\mathbf{s}}_0$ will be used to predict the reasoning result of the condition group. If the condition group is partially satisfied, we use the rest of vectors, $\hat{\mathbf{s}}_1, \ldots, \hat{\mathbf{s}}_n$, to identify unsatisfied conditions. We compute loss on both reasoning and unsatisfied condition predictions. Let $\mathbb{I}_r$ and $\mathbb{I}_c$ be the one-hot labels for the two tasks.

$$\hat{\mathbf{s}}_0, \hat{\mathbf{s}}_1, \ldots, \hat{\mathbf{s}}_n = \text{Reason}(\mathbf{s}_0, \mathbf{s}_1, \ldots, \mathbf{s}_n)$$

$$l_{\text{reason}} = \text{softmax\_cross\_entropy}(\mathbf{W}_l^T \hat{\mathbf{s}}_0, \mathbb{I}_r)$$

$$l_{\text{cond}} = \text{softmax\_cross\_entropy}(\mathbf{W}_c^T \hat{\mathbf{s}}_i, \mathbb{I}_c)$$

As discussed above (§3.1), the reasoning results of condition groups have three possible outcomes: "satisfied", "partially satisfied", and "contradicted". We merge the first two into one label "satisfied", and differentiate them by whether unsatisfied conditions exist, i.e. $r \in \{\text{satisfied}, \text{contradicted}\}$ and its one-hot label $\mathbb{I}_r \in \{0, 1\}^2$.[4]

Labels for conditions are "entailed", "contradicted", "not mentioned", "implied", and "unsatisfied", i.e. $\mathbb{I}_c \in \{0, 1\}^5$. The first three labels are as they are named. The label "implied" means a condition is implied by other conditions in the condition group. For example, if one of the conditions in a disjunction group has been satisfied, the rest of conditions are "implied". The class "unsatisfied" means it is an unsatisfied condition which must be returned together with the predicted answer. The labels may not apply to all datasets, e.g. ConditionalQA (Sun et al., 2021a) only annotates two labels "unsatisfied" vs. others, we will make changes to the loss function accordingly.

**Loss Function** We jointly train the entailment module and reasoning module. The final loss function is the sum of the answer loss $l_{\text{reason}}$ and the condition entailment loss $l_{\text{cond}}$. If the answers contain text spans, we jointly train the decoding module $l_{\text{decode}}$ as well.

$$l = l_{\text{reason}} + l_{\text{cond}}$$

$$l = l_{\text{reason}} + l_{\text{cond}} + l_{\text{decode}}$$

### 3.3 FINETUNE PRETRAINED CHECKPOINTS

The entailment module and decoding module (if adopted) load pretrained LM checkpoints, e.g. T5 (Raffel et al., 2019) and BART (Lewis et al., 2019). The pretrained parameters are loaded for the entailment module and then finetuned for downstream tasks. The reasoning module is randomly initialized and jointly trained with other modules. The number of Transformer layers in the reasoning module is a hyper-parameter. We choose the number of layers $l = 3$ or $l = 4$. Please see §4.1 for ablation study on the number of Transformer layers for the reasoning task. The decoding module is also finetuned. If a decoding module is needed, we will initialize the entailment and decoding module from the same pretrained checkpoint.

## 4 EXPERIMENTS

We experiment with T-Reasoner on a synthetic dataset, CondNLI, and two benchmark QA datasets, ConditionalQA (Sun et al., 2021a) and ShARC (Saeidi et al., 2018), for scenario-based QA task.

### 4.1 CONDNLI

**Dataset** The synthetic CondNLI dataset is derived from an existing Natural Language Inference (NLI) dataset, MultiNLI (Williams et al., 2018). An original NLI example contains a premise and a hypothesis, and a label indicating whether the premise is entailed or contradicted by the hypothesis. We treat premises in NLI examples as conditions and hypotheses as facts provided in user scenarios.

---

[4]Some tasks have an additional class "irrelevant" because some questions in the dataset are not relevant to the provided passages, i.e. $\mathbb{I}_r \in \{0, 1\}^3$.

| Context: | If all: [ |
|---|---|
| |     *"Aged 59 1/2 or older"*, |
| |     *"Employed for two years"* |
| | ] then *"Get at least $60 a week"*. |
| | If any: [ |
| |     not *"Has two children"*, |
| |     *"Has not applied before."* |
| | ] then *"Waive the application fees"*. |
| Question: | Is *"Eligible for $60 a week"* correct? |
| Scenario: | [*"65 years old"*, *"Rejected last year"*] |
| Answer: | Yes, [*"Employed for two years"*] |

Table 1: An example in CondNLI. The answer is "Yes" with unsatisfied conditions [*"Employed for two years"*].

|  | Ans (acc) | Conds (F1) |
|---|---|---|
| T5 (w/ FiD) | 85.6 | 80.4 |
| ETC | 91.4 | 82.7 |
| T-Reasoner | 95.0 | 91.3 |

Table 2: Experiment results on the synthetic CondNLI dataset in answer accuracy (Ans) and conditions F1 (Conds).

|  | 6 | 8 | 10 | 15 | 20 |
|---|---|---|---|---|---|
| T5 (w/ FiD) | 85.6 | 85.2 | 85.3 | 82.2 | 74.1 |
| T-Reasoner | 95.0 | 94.3 | 94.5 | 92.8 | 91.7 |

Table 3: Experiment results in generalizing to more conditions at inference time.

An example is shown in Table 1. The example contains four conditions, among which "Aged 59 1/2 or older" and "Employed for two years" belong to a condition group under a logical reasoning type "all", indicating that both conditions have to be satisfied in order to "Get at least $60 a week". The answer statement, e.g. "Get at least $60 a week", also comes from NLI examples. We treat the premise of an NLI example as an answer statement and the corresponding hypothesis as the question, e.g. is "Eligible for $60 a week" correct? In addition to the condition group and the answer statement that is relevant to the question, we add a few more condition groups as distractors to make the constructed dataset more challenging. Please see Appendix A for more information in dataset construction.

**Baselines** Previous work (Clark et al., 2020b) showed that pretrained Transformer-based Language Models, e.g. RoBERTa (Liu et al., 2019), have the ability to reason over multiple conditions to answer a reasoning question in the deductive reasoning setting, e.g. "if A and B then C" with facts on both conditions A and B provided. However, examples in CondNLI are usually longer and won't fit into RoBERTa's memory. Equivalently, we experiment with two other language models, T5 (Raffel et al., 2019) (with the FiD strategy (Izacard & Grave, 2020) to adapt to longer input) and ETC (Ainslie et al., 2020), on the CondNLI dataset.[5] In ETC, we use the global tokens to predict unsatisfied conditions. In T5, To simplify the generation task, we assign an id to each condition and let FiD generate unsatisfied condition ids. We also compare T-Reasoner with T5 on inputs that contains more conditions to test their generalization ability.

**Results** The experiment results are shown in Table 2. We measure both the accuracy of label prediction and the F1 of unsatisfied conditions. The results show that T-Reasoner performs significantly better than pretrained LMs, T5 and ETC, in both predicting correct answers (Ans) and unsatisfied conditions (Conds) on CondNLI. We additionally test T-Reasoner's ability in generalizing to more conditions. We train T-Reasoner on templates with 6 conditions or fewer and test it on the examples with more than 6 conditions. Figure 3 (Left) shows the change of performance in both label classification and unsatisfied condition prediction tasks as the number of conditions increase. We observe some decrease in performance in both tasks, but it is still reasonable with 20 conditions. Furthermore, we experiment with different numbers of layers in the reasoning module (Right). The Transformer-based reasoning module needs at least 3 layers for the reasoning task, especially for predicting unsatisfied conditions.

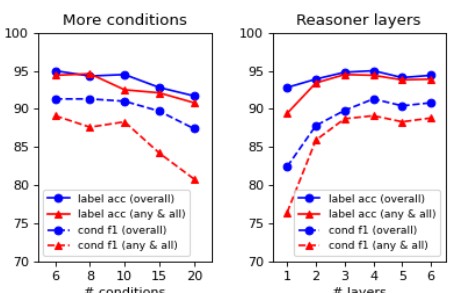

Figure 3: Left: Generalization results of reasoning over more conditions. Right: Results on the ablated model with different numbers of Transformer layers in the reasoning module. We report both label accuracy and F1 of unsatisfied conditions. "any & all" indicates the subset of validation data in which conditions interact each other under the relationship of "any" and "all".

## 4.2 SHARC

**Dataset** In the second experiment, we run T-Reasoner on a real scenario-based QA dataset, ShARC (Saeidi et al., 2018), that has complex passages and many conditions. An example in ShARC contains

---

[5]Examples in CondNLI exceeds the limit of 512 tokens in RoBERTa.

|  | Decision (micro / macro) | Question (BLEU1 / 4) |
|---|---|---|
| CM | 61.9 / 68.9 | 54.4 / 34.4 |
| BERTQA | 63.6 / 70.8 | 46.2 / 36.3 |
| UcraNet | 65.1 / 71.2 | 60.5 / 46.1 |
| Bison | 66.9 / 71.6 | 58.8 / 44.3 |
| E3 | 67.7 / 73.3 | 54.1 / 38.7 |
| EMT | 69.1 / 74.6 | 63.9 / 49.5 |
| DISCERN | 73.2 / 78.3 | 64.0 / 49.1 |
| DGM | 77.4 / 81.2 | 63.3 / 48.4 |
| T-Reasoner | **80.4 / 83.9** | **71.5 / 58.0** |

Table 4: Experimental results on the ShARC dataset. Numbers for the baseline models (Saeidi et al., 2018; Zhong & Zettlemoyer, 2019; Verma et al., 2020; Lawrence et al., 2019; Gao et al., 2020a;b; Ouyang et al., 2020) are borrowed from Ouyang et al. (2020).

|  | Decision (micro / macro) | Question (BLEU1 / 4) | Condition (F1) |
|---|---|---|---|
| T5 | 63.7 / 68.2 | 57.3 / 48.2 | 44.0 |
| DISCERN | 74.9 / 79.8 | 65.7 / 52.4 | 55.3 |
| DGM | 78.6 / 82.2 | **71.8 / 60.2** | 57.8 |
| T-Reasoner | **79.8 / 83.5** | **71.7 / 60.4** | **69.2** |

Table 5: Results on the ShARC dataset (dev) compared to the baselines. The Condition (F1) numbers are from open-sourced codes (Gao et al., 2020b; Ouyang et al., 2020).

| # conditions | 1 | 2 | 3 | 4 |
|---|---|---|---|---|
| DGM | 90.4 | 70.3 | 80.0 | 73.4 |
| T-Reasoner | 90.3 | 72.7 | 80.6 | 75.2 |
| *diff* | -0.1 | 2.4 | 0.6 | 1.8 |

Table 6: Ablated results on ShARC on answer accuracy vs. number of conditions. Numbers of DGM Ouyang et al. (2020) are obtained from open-sourced codes.

a passage, a user question, and a user scenario which is expressed in a conversation history between a user and a machine. A model is expected to find an answer to the user's question, or raise a clarification question for the unsatisfied conditions. Answers in this dataset are restricted to one of the following labels: "yes", "no", "inquire", and "irrelevant". The first three labels are equivalent to "satisfied", "contradicted", and "partially satisfied". "irrelvant" is a new label that should be predicted if the conversation history and the question are irrelevant to the provided passage. This task of predicting answers is called "Decision Making" in their original ShARC paper (Saeidi et al., 2018) and evaluated as micro and macro accuracy. In addition to the "Decision Making" task, they consider another task "Question Generation" which is equivalent to predicting unsatisfied condition in T-Reasoner,[6] evaluated with BLEU 1 and BLEU 4 scores. Compared to CondNLI, where conditions and their relationship are clearly mentioned in the context, conditions are embedded in the context in ShARC examples, e.g. Figure 1. Please see Appendix C for more information in data preparation.

**Baselines and Results** We compare T-Reasoner to several strong baseline models, including the previous state-of-the-art models, DISCERN (Gao et al., 2020b) and DGM (Ouyang et al., 2020). Different from the baseline models, which use pipeline systems to separately predict answer labels and unsatisfied conditions, T-Reasoner performs the two tasks jointly. The results are shown in Table 4. T-Reasoner outperforms the previous baselines by 3 points on the "Decision Making" task and more than 8 points on the "Question Generation" task. T-Reasoner also significantly outperforms other baseline models (Saeidi et al., 2018; Zhong & Zettlemoyer, 2019; Verma et al., 2020; Lawrence et al., 2019; Gao et al., 2020a;b; Ouyang et al., 2020).

**Ablation: Condition Accuracy** One problem with the ShARC Question Generation task is that only one of the unsatisfied conditions is annotated, even though multiple unsatisfied conditions exist. To further evaluate T-Reasoner's performance in predicting all unsatisfied conditions, we manually annotate the logical operations in 20 contexts that have more than one condition (857 data total),[7] and use the annotated logical operations to find all unsatisfied conditions. We report the F1 of the predicted unsatisfied conditions (see Table 5). Compared to the baselines (Gao et al., 2020b; Ouyang et al., 2020), T-Reasoner improves the F1 by 11.4 points.

**Ablation: Label Accuracy v.s. Conditions** We additionally measure the accuracy versus the number of conditions in the context. Results in Table 6 show that the improvement in T-Reasoner's performance over the previous state-of-the-art model (DGM) mostly comes from questions that have more than one condition.

## 4.3 CONDITIONALQA

**Dataset** In the third experiment, we run T-Reasoner on ConditionalQA (Sun et al., 2021a), which contains longer context (documents), more conditions and more complex relationship between

---

[6]Unsatisfied conditions are then paraphrased into questions, e.g. "Aged 59 1/2 or older" is paraphrased to "Are you aged 59 1/2 or older?"

[7]Each context in ShARC has 32.9 data on average.

| | Yes / No | | Extractive | | Conditional | | Overall | |
|---|---|---|---|---|---|---|---|---|
| | EM / F1 | w/ conds | EM / F1 | w/ conds | EM / F1 | w/ conds | EM / F1 | w/ conds |
| majority | 62.2 / 62.2 | 42.8 / 42.8 | – / – | – / – | – / – | – / – | – / – | – / – |
| ETC | 63.1 / 63.1 | 47.5 / 47.5 | 8.9 / 17.3 | 6.9 / 14.6 | 39.4 / 41.8 | 2.5 / 3.4 | 35.6 / 39.8 | 26.9 / 30.8 |
| DocHopper | 64.9 / 64.9 | 49.1 / 49.1 | 17.8 / 26.7 | 15.5 / 23.6 | 42.0 / 46.4 | 3.1 / 3.8 | 40.6 / 45.2 | 31.9 / 36.0 |
| T5 w/ FiD | 64.2 / 64.2 | 48.0 / 48.0 | 25.2 / 37.8 | 22.5 / 33.4 | 45.2 / 49.7 | 4.7 / 5.8 | 44.4 / 50.8 | 35.0 / 40.6 |
| T-Reasoner | **73.2 / 73.2** | **54.7 / 54.7** | **34.4 / 48.6** | **30.3 / 43.1** | **51.6 / 56.0** | **12.5 / 14.4** | **57.2 / 63.5** | **46.1 / 51.9** |

Table 7: Main results on ConditionalQA. The "EM/F1" columns reports the original EM/F1 metrics which only evaluate answers. The "w/ conds" is the conditional EM/F1 metric from ConditionalQA. See §4.3.

| | Answer (w/ conds) | Conditions (P / R / F1) |
|---|---|---|
| T5 w/ FiD | 3.2 / 4.6 | 98.3 / 2.6 / 2.7 |
| T-Reasoner | **10.6 / 12.2** | **34.4 / 40.4 / 37.8** |
| T5 w/ FiD (conditional only) | 6.8 / 7.4 | 12.8 / 63.0 / 21.3 |

Table 8: Ablated results on ConditionalQA in predicting unsatisfied conditions.

conditions. Furthermore, the ConditionalQA dataset contains a mixture of "Yes"/"No" questions and questions with free-form answers. Please see Appendix B for details on data preparation.

**Evaluation** As introduced in ConditionalQA (Sun et al., 2021a), predictions are evaluated in two sets of metrics: EM/F1 and conditional EM/F1. EM/F1 are the traditional metrics that measures the accuracy of predicted answer spans. *Conditional EM/F1* is a novel metric introduced by Sun et al. (2021a), that jointly measures the accuracy of answer spans and unsatisfied conditions. Please refer to the ConditionalQA paper (Sun et al., 2021a) for more information. Briefly, the conditional EM/F1 is the product of the original answer EM/F1 and the F1 of the predicted unsatisfied conditions. The conditional EM/F1 is 1.0 if and only if the predicted answer span is correct and all unsatisfied conditions are found. If there's no unsatisfied condition, the model should predict an empty set.

**Baselines and Results** We compare T-Reasoner with several strong baselines, including ETC (in a pipeline) (Ainslie et al., 2020), DocHopper (Sun et al., 2021b), and T5 (with FiD) (Izacard & Grave, 2020). The ETC pipeline first extracts possible answers from the context and then predict unsatisfied conditions independently. DocHopper is a multi-hop retrieval system that iteratively retrieves evidence which contains answers and unsatisfied conditions. T5 (w/ FiD) is a encoder-decoder model. We train T5 (w/ FiD) to generate answers followed by a list of unsatisfied conditions ids. The experimental results are presented in Table 7. T-Reasoner significantly outperforms the baselines in predicting answers and jointly predicting answers and unsatisfied conditions – a relative improvement of 148% (Conditional) and 27.8% (Overall) in conditional F1 (F1 w/ conds).

**Ablation: Condition Accuracy** Since there's not a metric that only measures the quality of predicted conditions, we additionally report the F1 of the predicted unsatisfied conditions (Table 2). The best baseline models, T5 (w/ FiD), rarely predicts any conditions. Even though we train T5 (w/ FiD) only on the subset of questions that have conditional answers to force it predict unsatisfied conditions, its performance slightly improves but is still much lower than T-Reasoner by 16.5 points in condition F1.

## 5 CONCLUSION

We study the problem of scenario-based QA in which questions are accompanied by incomplete scenarios and models are asked to find answers that are consistent with the provided user scenario. Models are further asked to identify unsatisfied conditions that are necessary for the predicted answers. We propose a system, T-Reasoner, that contains an entailment module to check whether a condition has been satisfied and a jointly trained reasoning module to verify the status of condition groups and predict unsatisfied conditions. T-Reasoner shows excellent reasoning ability, and can easily generalize to more conditions on a synthetic dataset CondNLI. Furthermore, T-Reasoner achieves state-of-the-art performance on two challenging scenario-based QA datasets ShARC (Saeidi et al., 2018) and ConditionalQA (Sun et al., 2021a).

## 6 ETHICAL STATEMENT

Experiments in this paper are performed on publicly available datasets for academic research purposes. No real user data is used in the experiments. Even though the proposed model could be applied to many real world problems to help users answer their questions, the accuracy of the proposed work is still limited and predictions may be misleading. Please carefully evaluate the performance before applying it to real problems.

## 7 REPRODUCIBILITY STATEMENT

Datasets and codes will be released upon the acceptance of this paper, including all scripts for constructing the proposed synthetic dataset and the preprocessing script for the two benchmark QA datasets. Models are trained on public available data. All results are reproducible.

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

| (Template) |
| --- |
| Context: If all [A, B], then U. |
| If any [not C, D], then V. |
| Facts: a, c, not d. |
| Question: Is u correct? |
| Label: Yes, if B |

| (Variables) | |
| --- | --- |
| A: Aged 59 1/2 or older. | a: Tom is 65 years old. |
| B: Employed for two years. | b: NOT_USED |
| C: Has two children | c: He has two sons. |
| D: Has not applied before. | not d: Rejected last year. |
| U: Get at least $60 a week | u: Eligible for $60 a week. |
| V: Waive the application fees | v: NOT_USED |

Table 9: An example of CondNLI. Variables $A, B, \ldots$ and $U, V, \ldots$ represent the conditions and premises. Variables $a, b, \ldots$ represent the known facts. $u$ is the question. Each pair of variables, e.g. $(A, a)$, is instantiated with an NLI example.

Michael J. Q. Zhang and Eunsol Choi. Situatedqa: Incorporating extra-linguistic contexts into qa, 2021.

Victor Zhong and Luke Zettlemoyer. E3: Entailment-driven extracting and editing for conversational machine reading. In *Proceedings of the 57th Annual Meeting of the Association for Computational Linguistics*, pp. 2310–2320, Florence, Italy, July 2019. Association for Computational Linguistics. doi: 10.18653/v1/P19-1223. URL https://aclanthology.org/P19-1223.

## A  CONDNLI DATASET CONSTRUCTION

We first construct templates for the CondNLI examples and then instantiate the variables in the template with real NLI examples.

**Construct Templates** We use capital letter $A, B, \ldots$ to represent conditions and lower-cased letters $a, b, \ldots$ to represent the corresponding facts. We use another few letters $X, Y, \ldots$ to represent the statements of conclusion, and lower-cased letters $x, y, \ldots$ to represent questions. Conditions are grouped together under a logical operator that specifies the relationship between conditions. For example, a logical operator "all" specifies that all conditions in the group must be satisfied in order to make the condition group satisfied. Here, we consider four types of logical operations to construct this synthetic dataset:

- "all": all conditions under this logical type should be satisfied in order to make the answer true.

- "any": only requires one of the conditions under the logical type "any" to be satisfied. It doesn't matter whether other conditions have been satisfied, contradicted, or not mentioned in the question.

- "required": This is a special case of "all" / "any" when there is only one condition. Conditions with the logical type "required" must be satisfied.

- "optional": Conditions have the type "optional" if they are not relevant to the question.

We pair a condition group with a conclusion statement and get a logical statement "If all $(A, B)$, then $X$". To challenge models' ability in identifying relevant conditions from context, we add a few distracting statement that leads to different conclusions, e.g. "If all (not $C$, $D$), then $Y$". An example of a context template is shown in Table 9.

Facts are constructed by randomly sampling a subset from all possible facts $\{a, b, \ldots\}$. A question is sampled from possible questions $\{x, y, \ldots\}$. We then compute the answer (and unsatisfied conditions if any) from the context, facts, and the question.

**Generate Examples** For a templates with variables $A, B, X, Y, \ldots, a, b, x, y, \ldots$, we instantiate the variables with NLI examples to get the real data. We use the premises of original NLI examples

| | |
|---|---|
| Answer labels (yes / no / negated yes / negated no / partially satisfied / irrelevant) | 100 / 100 / 100 / 100 / 100 / 50 |
| Logical types (any / all / required / optional) | 378 / 380 / 422 / 390 |
| # conditions in scenario (4 / 5 / 6) | 333 / 340 / 471 |

Table 10: CondNLI statistics. The table shows the statistics of 550 randomly generated examples. The answer labels "negated yes" and "negated no" mean negation exist in rule templates. We set the minimum number of conditions in scenarios to 4 to avoid overly simple example. Each example may contain multiple logical types.

for conditions and conclusions, i.e. capital letter variables, and the hypothesis for question and facts, i.e. lower-case variables. Note that sampling requires matching the entailment state of conditions, e.g. "not $d$" requires sampling from NLI examples that are labeled as "contradict".

We restrict the number of conditions in the context to 6 and randomly generate 65 distinct templates.[8] During training, we randomly pick a template and instantiate it with NLI examples to generate real training examples. This random generation process enables creating (almost) unlimited amount of training data. We randomly generate another 5000 examples for development and testing.

**Quality Assessment** Training and validation data in CondNLI are generated from NLI examples in the training and validation split of the MNLI dataset, respectively. This ensures that NLI examples used in validation are not exposed at training time. We control the generation process to ensure that the automatically generated data are balanced in terms of answer labels, logical types of interacting conditions, and number of conditions included in scenarios. Results are shown in Table 10. We additionally require scenarios must have at least 4 conditions to avoid overly simple examples.

We additionally measure the Jaccard distance between premises and hypotheses of the NLI examples used in constructing the CondNLI dataset. The token-level Jaccard distance is 27.2. Even though token-level overlap exists, a model still needs to understand the semantic relationship between premises and hypotheses to predict their entailment status.

## B  CONDITIONALQA EXPERIMENT DETAILS

An example in the ConditionalQA dataset provides a parsed web page as context. It also provides a question, and a user scenario that is relevant to the context. We prepend the user scenario to the question as input to the model.

The context in ConditionalQA is provided as a list of HTML elements. We treat each element at the leaf of the DOM tree as a condition $c_i$, and prepend all its parents (from its direct parent to the root) to get an expanded condition $s_i$.

Since we need the decoding module to generate answer spans, we initialize the model with T5, i.e. we use parameters from the encoder to initialize the entailment module, and use decoder to initialize the decoding module. The reasoning module is randomly initialized.

## C  SHARC EXPERIMENT DETAILS

Different from ConditionalQA, where each sentence in the context is treated as a condition, conditions in the ShARC dataset are shorter and are sometimes short phrases (sub-sentence). For example, the context "If you are a female Vietnam Veteran with a child who has a birth defect, you are eligible for ..." contains two conditions, "If you are a female Vietnam Veteran" and "with a child who has a birth defect".[9] In order to handle sub-sentence conditions, we follow the strategy proposed in two

---

[8]Restricting the number of conditions is only for the purpose of reducing training complexity. The experiment in Figure 3 (left) shows the model's capability of generalizing to more conditions.

[9]It is arguable that this could be generally treated as one condition, but it is treated as two conditions with the logical operator "all" in the ShARC dataset.

of the baseline models, DISCERN Gao et al. (2020b) and DGM Ouyang et al. (2020), that split a sentence into EDUs (Elementary Discourse Units) using a pretrained discourse segmentation model Li et al. (2018). The discourse segmentation model returns a list of sub-sentences, each considered as a condition.

While we could treat each condition independently as we did previously for other datasets, the segmented EDUs are different in that they are not full sentences and may not retain their semantic meaning. Thus, we consider using the full context (usually less than 512 tokens) as the contextual information for condition $c_i$, i.e. the expanded condition $s_i$ includes the full context, but the condition $c_i$ is highlighted using the special tokens `<CDT>` and `<\CDT>`.

We do not need the decoding module for the ShARC dataset, so we can safely discard it. We initialize the entailment module with ELECTRA (Clark et al., 2020a). The previous state-of-the-art baselines (Ouyang et al., 2020; Gao et al., 2020b) use ELECTRA to initialize their model. We use the same pretrained checkpoint to make a fair comparison.

For the question generation task, we use the same input $s$ as in decision making, except that we replace the prefix "condition:" with "unsatisfied condition:" for "unsatisfied" conditions. We fine-tune a T5 model for question generation.

## D   DATASET STATISTICS

Dataset statistics are shown in Table 11.

|  | Train | Dev | Test |
|---|---|---|---|
| ShARC | 15581 | 1622 | 5866 |
| ConditionalQA | 2338 | 285 | 804 |

Table 11: Dataset statistics.

