# OpenReview forum: "Scenario-based Question Answering with Interacting Contextual Properties"
_ICLR.cc/2023/Conference — ICLR 2023 poster_

### Official Review · Reviewer_nCC1 · 2022-10-25

**Confidence:** 4
**Correctness:** 4
**Technical Novelty And Significance:** 3
**Empirical Novelty And Significance:** 3
**Recommendation:** 6

**Clarity, Quality, Novelty And Reproducibility:**

The paper is mostly clearly written. The main ideas and the method are understandable from the text with some effort.

The work is of reasonable quality. The idea, even if somewhat straightforward, is will suited for the problem and represents a non-trivial advance for the specific problem.


**Strength And Weaknesses:**

Strengths

The paper addresses an understudied and challenging problem and provides a non-trivial empirical advance.

The proposed methods, while straightforward, are well motivated.

The writing is clear for the most part. See some presentation suggestions below.

Weaknesses

I wouldn’t say what follow are necessarily reasons to reject but are something that can be addressed to strengthen the contributions of this paper:

The description of the paper doesn’t provide a clear articulation of the key challenge or insight that is being addressed. It simply provides a description of the system and a demonstration that it provides empirical gains.

It would be useful to know if the specific design choices (e.g. having a summary vector for each condition and the additional transformer layers) are indeed necessary. I appreciate the information conveyed in Figure 3. It provides some evidence already. Will it be possible to add an experiment where the reasoner is removed completely? Instead, train a T5 model to do the two tasks directly. Also, It will be useful to analyze one baseline model to Figure 3 (left side). You have a similar analysis for the ShARC dataset in Table 5.



**Summary Of The Paper:**

The paper presents a method for answering questions about scenarios -- questions for which there isn’t a fixed answer but is varied depending on additional conditions that are unstated in the text.

Method: Given a scenario, a question, and a set of conditions (extracted from an input text) that are to be considered when answering the question, the method involves the following steps:

1. Use an "entailment" module to create entailment related representations of the information in the scenario against each of the conditions to be considered. Use this representation to make decisions about which conditions are satisfied, contradicted, or of unknown status.
2. The representations of the tokens in the conditions are also used to then generate the answer using a (standard) decoding module.

Model: The entailment module is a pretrained NLI encoder. The decoder is a pre-trained BART generator. The “reasoner” is a set of transformer layers that consume the aggregate representations of each condition from the entailment model. All components are fine-tuned end-to-end with three losses the specifically test for ability to produce correct answer, predict whether all conditions are satisfied, and predict for each individual condition whether it is satisfied, implied, contradicted, or not discussed etc.

Evaluations: The paper evaluates the proposed model on three datasets, one derived from the multi-NLI dataset and two Scenario QA datasets.

Key Contributions: The main contribution is in putting together the system with components that are tied to the steps involved in the process and a demonstration of the utility of this method on real datasets.

**Summary Of The Review:**

I think the paper presents a clear empirical advance using a simple but useful technique. This work would be a strong baseline for future methods to build and compare against.

---

> ### Author Response · Authors · 2022-11-16
> **Response to reviewer nCC1**
>
> Thank you so much for your valuable feedback. Please check out the new version of the paper, as labels of tables and figures have shifted.
>
> Results of your proposed experiments with a T5 model (without a reasoning module) were shown in Table 2 for CondNLI and Table 7 & 8 for ConditionalQA (named as T5 w/ FiD).The only difference with the original T5 model is that we adopt the Fusion-in-Decoder (FiD) trick from Izacard et al. [1] because inputs to models are usually long (> 2000 tokens) and won’t fit into the GPU memory. Experiments on ConditionalQA show that the performance of T5 w/ FiD is 12.8 points lower than T-Reasoner (44.4 vs. 57.2) in predicting answers, and is much worse in predicting unsatisfied conditions, i.e. 2.7 in F1 of conditions compared to 37.8 from T-Reasoner, 35.1 points absolute drop in performance.  We will clarify the presentation to make it clear that this experiment can also be viewed as an ablation.
>
> We additionally experiment with the original T5 (without FiD) on the ShARC dataset. Inputs from ShARC are relatively short, so they can fit into GPU memory. Results are added to Table 5. The performance of T5 is more than 10 points lower than T-Reasoner in predicting both answers and missing conditions.
>
> We update our "Introduction" section to clearly reflect the challenges of the proposed task posed on existing pretrained LMs. Specifically, we explain that the plain T5 model does not perform well as inputs become more complex, i.e. as the number of conditions increases and conditions become more complex. We conducted additional experiments (results presented in the new Table 3) to support this claim – accuracy of T5 drops by 11.5 points as the number of conditions increase from 6 to 20. Furthermore, we experiment with reducing the semantic complexity of conditions in the CondNLI dataset by replacing them with capital letters, e.g. “if you satisfy all A, B, C, then X”. The performance of T5 increases from 85.6 to 98.8. This shows that plain Transformer models could solve simple reasoning problems, but becomes less effective as the complexity of inputs increases. Therefore, additional modeling is required, e.g. by adopting a dedicated reasoning module in this paper, to explicitly model the logical interaction between conditions.
>
> [1] Gautier Izacard, Edouard Grave “Leveraging Passage Retrieval with Generative Models for Open Domain Question Answering”
>
> ===============
>
> Please let us know whether our response has clarified any of your concerns. If not, please let us know and we can further revise the paper or clarify any concerns.

---

> ### Author Response · Authors · 2022-11-30
> **Follow up**
>
> Hi reviewer nCC1,
>
> Could you please let us know whether our response has clarified any of your concerns? If not, please let us know and we can further revise the paper or clarify any concerns.
>
> Thanks,
> Authors

---

### Official Review · Reviewer_yAth · 2022-10-25

**Confidence:** 4
**Correctness:** 3
**Technical Novelty And Significance:** 2
**Empirical Novelty And Significance:** 3
**Recommendation:** 6

**Clarity, Quality, Novelty And Reproducibility:**

The code will be available so the results should reproducible if the authors specify all hyper-parameters used.
The proposed model is quite novel and improves over the baselines for

**Strength And Weaknesses:**

Strengths:
- The proposed approach is quite interesting.
- The model outperforms the baselines on multiple datasets.


Weaknesses:
- It would be interesting to add a baseline where the entailment module is trained and evaluated separately (possibly through silver training data generated by generating the training labels using any existing entailment model.
- It would be good to conduct a qualitative analysis (with examples) and a thorough error analysis showing how each component performs, how errors propagating from one module affects other modules, etc.
- The write-up needs improvements since there are several typos (e.g., special special, "regardless the status" --> "regardless of the status", "These information" --> "this information".

**Summary Of The Paper:**

The paper proposes a model to tackle scenario-based question-answering to predict the answer to a question along with unsatisfied conditions for the given user scenario. The proposed model comprises 3 components, an entailment module (to identify the condition), a reasoning module (that decides whether or not the conditions have been satisfied) and a decoding module (that outputs the answer spans for free-form questions). The proposed model outperforms the baselines on several datasets.

**Summary Of The Review:**

I think the authors propose an interesting model for scenario-based question-answering and the proposed model outperforms the baselines on multiple datasets. However, the paper would benefit from a more through error analysis.

---

> ### Author Response · Authors · 2022-11-16
> **Response to reviewer yAth**
>
> Thanks for your valuable feedback. We conducted the experiment you suggested to further analyze the behavior of our model. We experimented with the synthetic CondNLI dataset because this is the only dataset where intermediate labels for entailment are available. To be specific, we add a classification layer on the output vector of the entailment module, and supervise it with the original entailment labels from MNLI. The loss for this intermediate supervision is added to the final loss. We also measure the classification accuracy as an additional metric. Results show that the accuracy of the intermediate entailment task is 94.5, compared to 94.7 from a T5 deliberately finetuned on MNLI. The final accuracy of answers and unsatisfied conditions (F1) is 95.2 and 92.1, compared to 95.0 and 91.3 from T-Reasoner (Table 2). It shows that even though intermediate supervision with gold labels are provided for the entailment module, the performance only improves marginally.
>
> We also run the suggested experiment on the ConditionalQA dataset. We use the T5 model finetuned on MNLI as a pre-learned entailment model to label the entailment status of conditions. We use the silver data to provide intermediate supervision for the entailment module (same as the CondNLI experiment discussed above). The accuracy of answers (EM) drops from 57.2 to 42.1, even worse than plain T5 w/ Fid without any intermediate supervision (44.4). We suspect that intermediate supervision with silver labels may hurt the model's performance if they are noisy.
>
> ==============
>
> Please let us know whether our response has clarified any of your concerns. If not, please let us know and we can further revise the paper or clarify any concerns.

---

> ### Author Response · Authors · 2022-11-30
> **Follow up**
>
> Hi reviewer yAth,
>
> Could you please let us know whether our response has clarified any of your concerns? If not, please let us know and we can further revise the paper or clarify any concerns.
>
> Thanks,
> Authors

---

### Official Review · Reviewer_nxRZ · 2022-10-26

**Confidence:** 4
**Correctness:** 3
**Technical Novelty And Significance:** 3
**Empirical Novelty And Significance:** 3
**Recommendation:** 6

**Clarity, Quality, Novelty And Reproducibility:**

The paper is mostly understandable, and the results support the idea of the paper. Code and data will be available upon acceptance.

**Strength And Weaknesses:**

Strength:
- The paper models a relevant scenario and shows some interesting advancements in an important direction for QA systems.
- T-reasoner is a good attempt to jointly model the full process.

Weaknesses:
- Limitations about the synthetic dataset are missing (see my comments below)

**Summary Of The Paper:**

The paper looks at QA when questions are asked in a given scenario, and they can be answered only if providing the model information about the scenario. In addition, such questions require a high level of reasoning, and thus the model should also be able to infer how conditions interact with each other and find correct answers that possibly satisfy all the conditions. To deal with these challenges, the paper introduces a new dataset derived from an existing one (i.e., MultiNLI), and the T-reasoner that contains an entailment module to check if conditions are satisfied by the scenario, a decoding module to identify eligible answers, and finally a reasoning module. Results on the synthetic dataset proposed in this paper show that it outperforms other SOTA models.

**Summary Of The Review:**

Overall the paper clearly introduces the problem, and the modeling part is well supported with evidence.
The weakest part is the dataset introduced in this work, which is supposed to contribute to the paper. The dataset is derived from an existing dataset using some heuristics, but no data quality assessments are provided in the paper. For example, I'd be curious to see what the overlap between questions and scenario/constraints is, as it seems possible that overlapping plays an essential role in solving this dataset. I'd also highlight what the differences with other datasets are and why a new dataset is required to model this problem.

---

> ### Author Response · Authors · 2022-11-16
> **Response to review nxRZ**
>
> Thank you for your suggestion.
>
> We have included more discussions about the new dataset CondNLI in Appendix A in the revised version of the paper, including a more comprehensive discussion of the construction process of the synthetic data and the statistics of the dataset (e.g. distribution of logical operations and labels). To summarize, we enforce: (1) an equal distribution of positive and negative examples, (2) a uniform distribution of logical operations between conditions, e.g. “any”, and “all”, and (3) a balanced set of answers with or without conditions.
>
> To clarify your question on overlap between questions and scenario/constraints, conditions and scenarios in CondNLI (originally from MNLI) are much more complex than the ones used in the example in Table 1, which was only used for illustration purposes. A real example from MNLI, for example, contains a premise “Policies that provide insurance over the insured's entire life and the proceeds (face amount) are paid only upon death of the insured” and a hypothesis “Whole life policies are a type of life insurance that only cover the insured person until retirement from the workforce”. The average token-level Jaccard distance between premises and hypotheses in MNLI is only 27.2. Even though token-level overlap exists, a model still needs to understand the semantic relationship between premises and hypotheses to predict their entailment status.
>
> Very limited resources are available in studying the task of reasoning with logically interacting conditions. We expect the new dataset, CondNLI, to be a simple and easy-to-use benchmark. Compared to other datasets used in this paper, ShARC and ConditionalQA,  CondNLI is fully controllable in many ways, e.g. the size of the dataset, the distribution of labels, the number of conditions, types of logical relationship between conditions, etc. One may also consider obtaining intermediate labels for training, e.g. entailment status of each condition (an ablated experiment suggested by reviewer yAth), which are not available in ShARC and ConditionalQA. We will open-source our codes and encourage people to use it creatively for their own projects.
>
> ============
>
> Please let us know whether our response has clarified any of your concerns. If not, please let us know and we can further revise the paper or clarify any concerns.

---

> ### Author Response · Authors · 2022-11-30
> **Followup**
>
> Hi reviewer nxRZ,
>
> Could you please let us know whether our response has clarified any of your concerns? If not, please let us know and we can further revise the paper or clarify any concerns.
>
> Thanks,
> Authors

---

### Decision · Program_Chairs · 2023-01-20

**Decision:**

Accept: poster

**Justification For Why Not Higher Score:**

As one reviewer said, "The main contribution is in putting together the system with components that are tied to the steps involved in the process." I think the topic of the paper would be limited to only a subset of the ICLR audience.

**Justification For Why Not Lower Score:**

I think it is worth accepting the paper given the quality of the work and how the authors addressed reviewers' comments during the rebuttal.

**Metareview: Summary, Strengths And Weaknesses:**

This paper studies the problem of scenario-based question answering, where the system needs to output possible answers based on a user’s scenario, as well as unsatisfied conditions. The paper proposed a sophisticated 3-staged approach to tackle this problem, outperforming several SOTA approaches on multiple datasets.

All the reviewers give a weak acceptance of this paper, think the problem is new/interesting, and appreciate the modeling approach for this challenging task.

Some reviewers questioned the quality of the constructed dataset and missing ablation studies. The authors did a good job clarifying these issues and adding new results in the response phase. Hence, I recommend acceptance of the paper.


**Note From Pc:**

if the above contains the word "oral" or "spotlight" please see: "oral" presentation means -> notable-top-5% and "spotlight" means -> notable-top-25%. As stated in our emails, we are disassociating presentation type from AC recommendations